# Translating Health Coaching Training into Clinical Practice

**DOI:** 10.3390/ijerph192316075

**Published:** 2022-12-01

**Authors:** Anna McGlynn, Cathy O’Callaghan, Brendon McDougall, Julie Osborne, Ben Harris-Roxas

**Affiliations:** 1Population and Community Health, South Eastern Sydney Local Health District, Sydney 2050, Australia; 2Centre for Primary Health Care and Equity, University of New South Wales, Sydney 2052, Australia; 3Primary and Community Health, South Western Sydney Local Health District, Sydney 2170, Australia; 4School of Population Health, University of New South Wales, Sydney 2052, Australia

**Keywords:** health coaching training, chronic disease, self-management

## Abstract

Health coaching can benefit people with managing chronic conditions. It considers people’s motivations, is person-centred and has the capacity to promote healthy lifestyles and address chronic disease risk factors. However, how health coaching training is translated into routine clinical practice at unit and service levels has been under explored. A metropolitan local health district in Sydney, Australia provided coaching training to health professionals, but the extent to which coaching skills were translated into clinical practice was unknown. A redesign methodology was used to identify barriers and facilitators for training-to-practice translation. Survey and workshop findings indicated that participants were satisfied with the coaching training but found it challenging to apply in clinical practice. Identified opportunities to support the application of health coaching were tailored practical training, post training support, and consensus on the definition of health coaching. Solutions were to develop an internal practical training program, use consistent terminology, and embed organisational support. Adoption of health coaching needs to occur on three levels; individual, workplace and organisation to ensure effective health care delivery. This case study demonstrates the importance of evaluation and diagnostics of contextual barriers and enablers to inform translation into practice.

## 1. Introduction

Chronic conditions are the leading causes of death and disability worldwide with disease rates continuing to escalate [1,2,3]. The successful management of these conditions depends on an integrated health system and patient self-management [4]. The World Health Organisation’s framework on providing integrated people centred care challenges health services to address the root causes which impact across the major chronic conditions [5]. Strategies include empowering and engaging people and communities; strengthening governance and accountability; reorienting the model of care; coordinating services within and across sectors; and creating an enabling environment. Best practice principles of integrated care outline broad considerations from the micro to macro perspective at the individual, organisational and systemic levels [6]. A person-centred approach is crucial to establishing integrated systems coordinated around people’s needs, that deliver the required care in the right place, at the right time and for the right cost [7]. The culture change needed to achieve person-centred care requires a whole of system approach and structural support [8,9]. Training clinicians to develop skills in health coaching supports work towards this transition.

Health coaching is recognised as an effective intervention that supports people to self-manage their chronic conditions as part of a health management program. It is reported to improve behaviour change and quality of life [10,11,12,13]; reduce costs for the healthcare system and the consumer; and is one of many tools that health services can use to improve health outcomes for patients. While there are overall reported benefits, the actual assessment of evidence for health coaching on physical outcomes is mixed due to the varied components and ways in which coaching is understood, delivered, and assessed [11,14].

Health coaching is conceptualised in varying ways. However, there is broad agreement it is ‘a patient-centred approach to goal-setting, active learning and self-management that guides, empowers and motivates an individual to change their behaviour’ [11,15]. The literature has no consensus on the most effective ways to train clinicians to deliver health coaching. Often studies do not provide enough detail on program content, training approaches and competencies to enable their replication [11,14].

Stand-alone training programs to enhance clinicians’ knowledge and skills are not sufficient to ensure desired changes are embedded into clinical practice. A review of health coaching training suggests that programs should contain tailored practice, incorporate post training assessment of competency and skills, and provide opportunities for ongoing support to motivate clinicians and enhance their competencies [14]. Organisational support structures should also be in place [16]. One Australian study reported that only 50% of health coaching trained participants went on to implement health coaching within a rural community health setting; the greatest reported limitations were time, knowing how to implement, and clinician confidence [17].

Improving translation of coaching training into clinical practice has flow-on benefits for patients. It is estimated that translating evidence into regular clinical practice takes over 20 years [18]. Yet, there is a gap in the literature of supportive strategies, assessments and considerations required to embed health coaching training into practice at the individual, organisational and systemic levels [6]. A thorough diagnostic exploration identifies barriers to translation and areas for targeted improvements.

The context of this case study is a metropolitan regional health organisation (local health district) in the state of New South Wales, Australia. The district implemented a training program to provide clinicians with skills in health coaching as part of a broader, state-wide chronic disease management initiative to enhance integrated care, support collaborative care planning and help patients self-manage towards better health. This program aimed to equip clinicians with skills to have coaching conversations with patients and support self-management within the patient’s usual care and service arrangements. It was delivered at no cost to participants and was accessed by 721 individuals from 2012 to 2017.

Clinicians providing services to patients with chronic diseases were initially targeted to attend training. This cohort was soon expanded to include Medical, Nursing, Allied Health and Administrative staff across acute, outpatient and community health settings. Training was also offered to Primary Care General Practitioners and Practice Nurses within the district’s geographical boundaries.

Training included an initial 2-day course which was delivered by an external training provider, with participants encouraged to attend a follow-up workshop 6 months post completion of initial training. Further opportunities were made available through the external training provider, including a 1-day managers’ workshop, 1-day Practice Nurse workshop, and 1-day workshop on providing health coaching in a group setting.

Training and workshops were predominantly theoretical with limited opportunities for participants to practice and apply their skills in a supported learning environment. The coaching model taught followed a systematic method of identifying and addressing individuals’ barriers to change. The training organisation designed and held the copyright for this model.

Five years after implementation of the training program, two of the manuscript authors noticed many clinicians were still finding it difficult to embed coaching conversations into their usual care.

## 2. Methods

To understand how health coaching training was translated and applied into clinical practice, a project team was formed with members from the district’s Integrated Care Unit and the local Primary Health Network (PHN). The team undertook a systematic redesign approach to identify barriers and enablers to implementing health coaching in clinical practice. The redesign methodology used includes the phases of initiation, diagnostics, solutions, implementation and sustainability. Redesign methodologies are applied to improve processes in health care settings in Australia and internationally [19,20]. This approach enabled the team to identify the root causes of issues impacting on service delivery, then develop and implement sustainable change processes [21].

The diagnostics phase of the project included retrospective surveys of health coaching trainees from the previous five years; health professional and consumer focus groups; healthcare professional interviews; data analysis; and root cause analysis. A Human Research Ethics Committee determined this was a quality improvement project (SWSLHD HREC Ref 2021/ETH00130).

An online survey was created and sent to the 429 training participants for whom the project team had current email addresses. The survey aimed to find out whether participants had implemented health coaching into patient care including how frequently coaching was used, barriers and facilitators to implementation, as well as clinicians’ culture and beliefs regarding health coaching.

## 3. Results

One hundred and forty-nine survey responses were received, representing a 34.7% return rate. 78% of responses were received from current district employees, with the remainder from local Primary Care and former district employees (Figure 1).

Allied Health professionals provided 54% of responses, followed by Nursing at 21%. The remainder were from Primary Care clinicians and Healthcare Managers. Reponses were received from a variety of settings, including Community Health (33%), Outpatient Services (20%), Inpatient Care (16%), Primary Care (9%), and other settings (22%).

The survey revealed high clinician acceptability for the use of health coaching, with 93% of respondents reporting they would continue to support health coaching. Furthermore, 85% believed health coaching is a legitimate part of their role as a clinician.

98% of respondents believed there are patient benefits to participating in health coaching. 97% believed there are clinician benefits and 79% valued the effect coaching can have on their own work. However, 54% indicated they were able to implement and regularly deliver coaching.

The survey results provided some initial insights into why the health coaching training was not translated into practice, with only 22% of respondents agreeing there were sufficient resources available to support health coaching, and 27% agreeing they received sufficient training to implement health coaching.

The qualitative data from interviews, focus groups and workshop consultations were analysed using a thematic approach [22]. This identified three key issues (Table 1); insufficient tailored training on health coaching and how to implement; insufficient post-training support and resources; lack of understanding, consensus and knowledge of health coaching.

## 4. Discussion

The low rate of translation of health coaching skills into clinical practice led the district to further explore the situation and redesign their approach to health coaching training. Common barriers affecting the uptake of health evidence into practice can include approach feasibility; professional attributes; patient attributes; professionals’ social context; organisational support; and economic and political context [23]. In this study, the factors impacting on the translation of health coaching existed at three levels: individual; workplace; and organisational. This resonates with the Rainbow Model of Integrated Care, where integrated care interventions are more likely to be successful and sustainable when they exist across micro, meso and macro levels [6].

### 4.1. Individual Level Factors Impacting Successful Translation

Factors affecting implementation occurred on the individual clinician level in relation to them having sufficient time and professional capacity to incorporate a separate model of health coaching into practice.

Several clinicians reported that health coaching as it was understood was time-consuming to implement. This appeared to relate to the underlying perception that health coaching was an add-on to their usual discipline-specific treatment methods, rather than a tool that supported regular practice. One reason for this related to the rigid and complex model taught, which reinforced a belief that health coaching was a separate process. Clinicians thought they needed to implement a long stepped process which would take significant time (not just 1 or 2 steps that would be of most value to the client and less time consuming). This created a disparity between the perceived and actual time required to implement. Clinicians were found to be less likely to implement health coaching into usual practice when they did not understand its benefits or application in the workplace. The training delivered did not adequately cover implementation of this skill into practice and the content was not specifically tailored to the clinician’s context, clinical area or patient type [14].

The support and resource requirements for training participants had not been designed specifically to those who were being trained in this district. Participants felt that post-training support structures were not adequate. The literature strongly supports providing practical support after initial training as an effective approach to ensure successful implementation [14].

Clinician autonomy was found to be an enabler for providing health coaching. The importance of autonomy has been discussed in other research, including clinicians having the ability to organise follow up appointments with patients to discuss progress with health goals [24]. Autonomy is also dependent on the nature and status of the profession and organisational/managerial support. The characteristics of supportive workplaces included allowing innovation, encouraging trialling and testing of new processes and ideas, and providing flexibility with scheduling patients.

### 4.2. Workplace Level Factors Impacting Successful Translation

Clinicians reported that their ability to implement health coaching was impacted by time schedules, managerial and peer support and quality improvement review processes within the workplace. Addressing these factors requires broader change at the service level.

The most frequently reported barrier to providing health coaching was a lack of time within professional and departmental daily schedules. Conversely, the organisation of time within the workplace was noted as an enabler by the clinicians consulted. Clinic structure, appointment scheduling and a tolerance for increased consultation times were found to support the uptake of health coaching. This was operationalised in several ways, such as having the workplace capacity to schedule longer appointment times, establishing dedicated appointment and clinic times for health coaching, and utilising any available spare appointment time with patients. Having a manageable workload also supported clinicians to spend the time required to implement health coaching as part of their usual practice. Rigid and long-standing assessment and treatment protocols were reported to be unconducive to providing health coaching. Coaching literature also outlines that flexibility in timing of appointments is needed as well as a manageable workload to enable implementation [25].

Support from colleagues, managers and the organisation were the most frequently cited enablers. However, less than half of the survey respondents felt that their managers adequately supported health coaching. Critical enablers included management support to attend the health coaching training and to practice health coaching in the workplace. Changing workplace processes to accommodate health coaching provided crucial reinforcement. Enabling factors for positive results include organisational, managerial, leader and peer support [14,16,17].

To provide a supportive environment, changes need to be fostered across the workplace. Establishing health coaching conversations as business as usual for the entire team was found to be a highly supportive approach. These services often attended training as a team, with management and peers then working together to embed the changes required. Research highlights the need for practical and emotional support for the people delivering coaching through group and team coaching and other staff facilitating coaching [24,26]. The results highlight the importance of engaging management at the service level to ensure that workplace culture, structures and processes sustain the delivery of health coaching.

Workplace processes that encouraged delivery of health coaching included regular service and role review, and a focus on quality improvement. These were further supported by documented assessment and treatment protocols, as well as booking processes where patients who could benefit from coaching are booked with clinicians trained in health coaching. Enablers for successful implementation include programs that are based on robust scientific evidence, informed by local data, fitting with local culture and staff preferences, and are supported by leadership, strong monitoring and feedback systems [27].

### 4.3. Organisation Level Factors Impacting Successful Translation

The clinicians consulted identified practices and structures that supported or hindered health coaching at the broader organisation level including funding structures, data reporting systems, and a shared understanding of coaching. Systemic organisational strategies such as these are harder to implement [6].

Issues were identified with the structure of the current health coaching training program and the organisation providing ongoing educational support and resources. Clinicians with ongoing access to peers knowledgeable in health coaching or where management and peers continued to work together to enhance coaching skills found this to be enabling. Those without access to ongoing training or support from subject matter experts noted that this was a barrier to implementation. Clinicians had limited knowledge of further health coaching education available or of advanced learning opportunities that addressed difficult coaching scenarios. These findings reinforce the translational literature, which indicates that training provided should include practical support around managing the change required to practice and apply skills in the workplace [14]. Important components in training include practice and observation, role play, and incorporating follow up sessions for further practice, support and ability to receive feedback on performance.

The state’s activity-based funding model incentivises health services based on the volume of care provided rather than the quality or patient outcomes [28,29]. Other factors such as the need to reduce patient waiting lists and meet daily demand can also force rigid and short appointment times. This discourages providing the more flexible appointment times that may be needed to deliver health coaching, especially for newly trained staff. Furthermore, clinicians reported that the forms and templates in the electronic medical record and communication systems were not set-up to include documentation or sharing of health coaching discussions, plans and outcomes.

There were large variations in the interpretation of what health coaching looked like when put into practice. For some participants, health coaching was considered to be a quick conversation with the patient about their goals and treatment while providing discipline related services (i.e., a physiotherapist informally talking to a patient while walking alongside them). Other participants interpreted health coaching to be a formal and structured sit-down conversation lasting an hour in duration. This variability also caused difficulties for the project team, as the identified enablers and barriers varied according to the type of coaching model clinicians aimed to implement. The literature lacked an agreed definition for Health Coaching in practice [11,15]. This posed issues for evaluating coaching implementation, and its subsequent effectiveness in changing behaviours and outcomes. Several of the clinicians consulted indicated that there was no clear vision for health coaching within the district, and minimal organisational support. These factors are both considered essential components for implementing coaching learnings into practice, optimal performance and ongoing sustainability [30].

It was acknowledged that the training program had a very limited practical component. Translational literature recommends that a structured and practical model is essential for translation of knowledge into practice [13]. The literature supports the need for supervision, follow up and constructive feedback in the workplace post-education as well as ongoing support for health professionals, and a receptive organisational context [14]. Many of these factors were not in place in this Local Health District and may have contributed to the difficulties in translation into practice.

## 5. Solutions/Enablers

The thorough diagnostic work on translating health coaching training into practice was essential to truly understand the supports clinicians needed. The redesign project team conducted a series of workshops to report the barriers and enablers identified from the survey back to the clinicians participating in the consultations. This process provided a forum for further discussion of the results and identification of a series of future actions. These actions were prioritised based on the project team’s ability to influence the issue and its likely impact on improving the translation of health coaching into clinical practice. The project team addressed the following three issues:(1)Content and structure of training tailored to context(2)Availability of post training support and resources(3)Variance in knowledge and understanding of health coaching

The team worked with the district’s Organisational Learning and Development (OD&L) Unit to develop and deliver an internally led health coaching program that addressed the identified needs of tailored and post training support, entitled Coaching for Better Health Outcomes (CFBHO). This ensured that clinicians trained in health coaching received the skills and supports required to successfully implement what they had learnt. Throughout the program design, there was a strong focus on tailoring the training to the clinicians’ local context and using a consistent definition of health coaching. Post-training support was built into the program design. The CFBHO program uses a solution focused and strengths-based approach to coaching, with the program delivered through cooperative and participative spaced learning. Changes were made at individual and workplace levels to translate training into practice.

A district definition of health coaching was developed and implemented across the organisation based on current evidence [11]. This agreed definition enabled a more consistent approach to the assessment of the effectiveness of this modality in changing health behaviours and outcomes [15], and will provide more accurate results for future evaluations.

These issues with translating health coaching into practice were addressed across the various levels as described in Table 2.

A pilot of the two-day workshop was held in April 2018 with the CFBHO program launched in July 2018. Throughout 2018, a total of 83 individuals attended the CFBHO two-day workshop. The CFBHO program will be thoroughly evaluated and described at a later date, however preliminary program outcomes indicated that CFBHO improved translation of health coaching training into clinical practice. Furthermore, a higher percentage of participants agree that there is sufficient training available to implement health coaching in their clinical roles, and that there are sufficient supports and resources to continue to implement and provide coaching.

While these results are preliminary, they are encouraging and support the development and delivery of health coaching programs that provide training and ongoing support for skills translation. This program appears to be successful as it considers the local context at the individual, workplace and system level and addressed local barriers and enablers.

While having an internal unit is not essential to enable health coaching translation, this did provide a consistent ongoing approach to understand the local context and provide appropriate supports.

## 6. Significance

This study is significant in describing an evaluation of barriers and enablers to translating health coaching training into practice in a metropolitan state-funded health service in Sydney, Australia. The focus on translation of health coaching training into practice was previously a gap in the literature. The strengths of this approach included a thorough diagnostics piece of work to truly understand the current state, and then determine the actions to better support implementation.

The survey and workshops that informed the redesign diagnostics, analysis and solutions only captured willing respondents who had participated in the health coaching training program. This may not necessarily reflect the enablers and barriers for all health coaching training participants, particularly those disinvested from health coaching. Therefore, it is possible that this group faces different and perhaps more significant barriers to implementing health coaching into practice.

Health care organisations should undertake diagnostic analysis of their health coaching training translation rates and publish the findings. This may help to provide further insight into the common barriers and solutions to assist areas without the resources to do this work. Now that the new program has been piloted and is shown to address previous barriers, it will be scaled up, evaluated and the outcomes published.

## 7. Lessons Learned

Health coaching training is an important aspect of person-centred chronic disease management, but it is understood in different ways.

Delivering health coaching training is not enough to ensure implementation into clinical practice, there needs to be ongoing support at clinician, workplace and organisational levels of the health system.

Considerations of contextual barriers and enablers need to occur when developing and delivering training

A thorough diagnostic exploration is an effective way to understand what the real issues are for clinicians trying to translate training into practice.

## 8. Conclusions

It is recognised that the culture change needed to achieve person-centred care requires a whole of system approach. Likewise, embedding health coaching into business as usual requires more than training clinicians, it requires having an overarching strategy or model that supports care to be delivered in a more person-centred way.

This project found there was limited evidence of the best way to achieve widespread use of coaching in an organisation. This makes it difficult for organisations to provide coaching training to their staff and know if the benefits are reaching their patients.

The barriers and enablers for translating health coaching training into practice occurred at three levels: patient/provider, workplace and organisational. For successful translation to occur, issues must be addressed at all three levels.

Delivering health coaching training as a once-off program was found to have its limitations. It is important to evaluate and address barriers to implementation. An education model that provides ongoing support and access to more advanced coaching techniques to build skill over time can lead to better translation rates.

A sustainable solution requires an organisational response that addresses clinician autonomy, time, support, compatibility with the existing workplace environment, and establishing a common understanding of health coaching across the organisation. Additional studies in this area could determine the broader applicability of the key factors identified for supporting the translation of health coaching education into practice.

## Figures and Tables

**Figure 1 ijerph-19-16075-f001:**
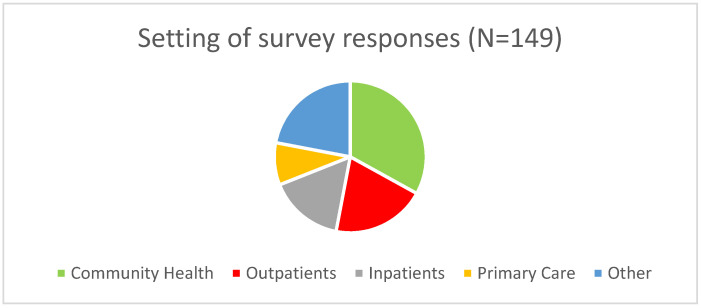
Survey responses received by area of work.

**Table 1 ijerph-19-16075-t001:** Translating health coaching into practice—key issues, root causes, impact and priority.

Key Issue	Root Cause	Impact	Priority
1. Insufficient tailored training on health coaching and how to implement	Health coaching training is not tailored to the healthcare providers context (such as their clinical area and patient type)	Providers find it difficult to apply training into practice which could impact on poor patient outcomes	Moderate
	There is limited promotion and knowledge of further education and learning opportunities	No participation in further learning to improve health coaching skills	High
	Training does not adequately cover how to implement health coaching into practice	Providers lack the confidence and skills to implement health coaching	Moderate
2. Insufficient post-training support and resources	There was no district staff, external contact or expert group dedicated to supporting providers of health coaching	Providers unable to seek support & advice, which meant a lack of health coaching being provided	Moderate
	Healthcare providers have not been asked what support and resources they require	Support and resource needs unknown	High
	Health coaching is generally viewed as an “add-on” to clinical care so does not become supported standard practice	Low adoption of health coaching	Moderate
3. Lack of understanding, consensus and knowledge of health coaching	There is no standard definition of health coaching used by the district, primary care and education providers	Variance in what providers consider ‘health coaching’	High
	The purpose and benefits of health coaching are not well promoted	Lack of support and provision of health coaching may impact on patient health outcomes	High
	The complex health coaching methodologies used by the current training provider led to a misunderstanding of what health coaching can be in its simplest form	Misconceptions existed about the amount of time health coaching takes and ease of health coaching provision	High

**Table 2 ijerph-19-16075-t002:** Barriers and solutions to translating health coaching in practice.

Barriers/Issues	Focus of Level Consideration	Enablers/Solutions
Rigid complex structure of training not tailored to local context (clinical area, patient type etc)	- Individual - Workplace	- Simplified flexible coaching structure applied - Training tailored to clinical area and patient types with a practical focus on how to implement
Post training support and resources	- Individual - Workplace	- Ongoing assessment of needs - Resources provided post training - Expert group dedicated to support - Cooperative approach to training and support provided by workplace managers
Coaching definition	- Organisational	- Standard definition used across organisation and in training program

## Data Availability

Not applicable.

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
