# Peer review of "Translating Health Coaching Training into Clinical Practice"

_ijerph, 2022, doi:10.3390/ijerph192316075_

Round 1

Reviewer 1 Report

Thank you for the opportunity to review this interesting article. It addresses a very important topic and well-written.

Below are my comments/suggestions:

1.      Authors may state at the introductory section what the study aim(s) and objective(s) are. This could be done at the end the introduction. When you read the manuscript, it is not very clear what the study aims to achieve.

2.      I see name of professional groups such as: nursing, allied health and administrative staff primary care general practitioners and practice nurses having initial capital letters. Any reason for this? Because I see these to be common nouns.

3.      Authors may consider expanding the methodology section. It appears they collected both quantitative and qualitative data. But it is not clear how the survey data from the 429 participants was analyzed. Also, what is the nature of the survey instrument? Inclusion of some description will help.

4.      If it is not the style of the journal, then I suggest the authors avoid beginning sentences with figures. For instance, 78% of responses were received ……………………… may be Seventy-eight per cent of responses ……………………..

Author Response

  1. Thanks for your kind feedback. We also believe this is an important topic for health care managers and policy makers to be aware of. The aims have now been made clear in the introduction, thanks for the suggestion.
  2. The names of professional groups have been changed to remove capital letters.
  3. I’ve expanded upon this and included your suggestions, thanks for the feedback.
  4. The sentences have been changed to avoid starting with a numerical figure.

Reviewer 2 Report

This manuscript describes a highly relevant and inspiring case study describing how the authors systematically followed up on unsatisfactory results from the implementation of a coaching program  

Background

The project is described  as a case-study, but although the structure and criteria of case studies are not as strict as other designs, I find that the aim of this study should be more explicit at the beginning (ex the end of the background). I also suggest it includes a plan on how to follow-up on the thesis that the implementation has failed. It may also contribute to a better understanding of how the case-study is presented and structured

Methods:

The authors have described the phases and the methods used to redesign the process, however it is unclear if it is based on an implementation framework (like e.g The Re-Aim Framework or CFIR framework).

As the response rate on the questionnaire was relatively low it could be relevant to know more about how questionnaires were developed and tested.

Results:

I think it would make more sense if the Solustions/Enablers were a part of the ‘Results’

And to benefit more from the experiences of the two types of coaching interventions implemented, I suggest that the authors describe the programs' content and differences more detailed.

Discussion:

In the discussion section I missed a more in-deep discussion of the meaning of theoretical learning versus more concrete training of lessons learned e.g by using theories of Albert Bandura, the evidens from self-efficacy studies or maybe studies using the Self-determination Theory (as is also describes the meaning of Autonomy)

Author Response

  1. Thanks for your kind feedback. The heading has been changed to Solutions as enablers are covered in results and discussion. This was not raised by the other reviewer and we are of the opinion that it makes more sense to discuss what all of the difficulties with translating are in the results section as this is what the focus of this study is.

We are hoping this article will inspire others to undertake this process themselves rather than just copy our solutions as that may not work in their context. Our colleagues intend to publish the new model for coaching once it’s out of the pilot phase, including what aspects have been changed to improve translation. For this reason so we don’t want to provide too much information on this at present.

  1. The discussion section has been amended to include a discussion of the importance of applying skills in the workplace to build self-efficacy and peer discussion of lessons learnt in a community of practice
  2. The ACI redesign methodology is what was used. It’s based on a few methodologies including Re-Aim. We have briefly mentioned redesign in lines 105-111.

More detail has been provided

  1. Added an extra sentence at the end of the abstract and introduction to make the aims clear.

Also added lines 157-160 to make it clear as to why we believe the original failed.

Round 2

Reviewer 2 Report

The authors have accommodated my input in a satisfactory manner